# Impacts of Climate Change in Baja California Winegrape Yield

**Marilina Hernandez Garcia** [1,*]**, María Cristina Garza-Lagler** [2]**, Tereza Cavazos** [3] **and Ileana Espejel** [1]

1 Oceanological Research Institute, Autonomous University of Baja California, Ensenada 22873, Mexico; ileana.espejel@uabc.edu.mx

2 Research Center in Food and Development, Hermosillo 83304, Mexico; maria.garza@ciad.mx

3 Department of Physical Oceanography, Ensenada Center for Scientific Research and Higher Education, CICESE, Ensenada 22860, Mexico; tcavazos@cicese.mx

\* Correspondence: marilina.hernandez@uabc.edu.mx

**Abstract:** We analyzed climate change scenarios and their possible impacts on winegrape yield in Baja California, the leading wine producer in Mexico. Linear regression models were used to predict the current yield based on climate and economic variables. Using future projections of the climate variables from two regional climate models (RegCM and RCA4), we evaluated the possible changes in yield for the Near Future (NF: 2021−2040) and Intermediate Future (IF: 2041−2060) periods under low (RCP2.6) and high (RCP8.5) greenhouse gas emissions scenarios. One regression model includes maximum and minimum temperatures (Tx and Tn) of the winegrape growing season and accumulated winter precipitation (Pre), and the other model also includes the real minimum wage and winegrape price to evaluate the operating cost paid by producers. The results show that the linear regression model with the climatic and economic variables explains 28% of the winegrape yield, and Tx and Tn had the greatest influence. The climate change scenarios show that during the winegrape growing season, these variables could increase more than 1 °C in the NF and more than 2 °C in the IF under the RCP8.5 scenario. These latter temperature changes could reduce the yield between 18% and 35% relative to the reference observed climate dataset (Livneh). However, winegrape yield is sensitive to economic factors, as the yield reduction increases at least 3% in all cases. Thus, adaptation strategies need to be implemented in the viticulture sector to reduce future impacts.

**Keywords:** winegrape; climate change; Baja California viticulture; global warming

## 1. Introduction

Baja California is the second largest producer of industrial winegrapes in Mexico after Zacatecas [1] and the only state in Mexico with a Mediterranean climate. According to the Mexican Vitiviniculture Council, about 75% of the national wine production [2] and 33.6% of the national winegrape production are concentrated in the state [3]. Viticulture activities contribute to the local economy and have attracted synergy with other sectors such as restaurants, hotels, and tourism. In 2021, the cultivation of winegrapes generated an economic benefit of 397 million MXN in Baja California [4], approximately 20 million USD. This production is related to an increase in wine houses; in 2000, there were eight wine-producing houses, whereas in 2015, there were 89, representing an increase in the wine supply of 1.113% in fifteen years and an annual growth of 74% [5].

With unfavorable climatic conditions (droughts and high temperatures during the growing season) and changes in the regional economy, yield and the agricultural areas destined to winegrape planting were considerably reduced in Baja California in the 1990s [4,6]. The total record (1982–2022) shows that the minimum area was observed in 2005–2006, but it has slowly increased in the last 15 years (Figure 1). In 1984, the area destined to winegrape production was 6039 ha, whereas in 2022, there was 33% less area [4]. Therefore, the highest regional yield was ~13.5 tons/ha in 1982, and in the last years, it has decreased 52% (Figure 1); a negative trend in yield has been observed since 1996, and there is no

sign of recovery. This trend may continue, as climate change is expected to produce more interannual uncertainty in the region [1].

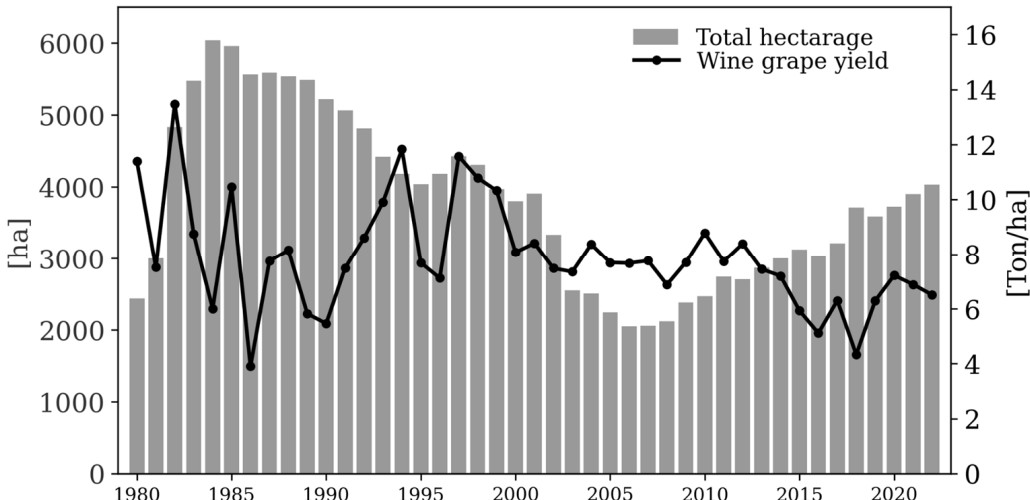

**Figure 1.** Total hectarage and winegrape yield in the Mediterranean region of Northwestern Baja California during 1980–2022. Data sources: [4].

Winegrape cultivation is highly sensitive to climate variations [7–11]. Temperature and precipitation are the main atmospheric variables for production and quality [8,12]. In recent decades, climate change effects have been evident in increasing temperatures at a global scale [13,14]. If the effects of social and economic problems are added, such as the market conditions in each winegrape-producing region [15–18], climate change becomes a great challenge for winegrape production [9]. It has been shown that in warmer winters and springs, the grape growing and harvest seasons are brought forward by 18 to 24 days [19]. In Baja California, this early onset of the growing season is also expected with global warming affecting grape phenology, which in turn may produce negative changes in the wine quality [1,20].

Climatic and economic variables are well correlated with agricultural production, including viticulture, e.g., [21–23], such as temperature, precipitation, crop yield, and economic factors [9,15]. Some studies have used a climate-related model to predict agricultural yield and the impacts of some climatic variables on the suitability regions for grape planting [24–26], and others have focused on assessing the impact on grape growth and, consequently, the effect on the yield [24,27–35].

In this work, we present an estimation of the climate change impacts on the winegrape yield in Baja California during 2021–2040 and 2041–2060 for two representative concentration pathways (RCP2.6 and RCP8.5) of greenhouse gas emissions or radiative forcing scenarios. The relationship between the grape yield with climatic and economic variables is determined with a linear regression model for a historical period (1981–2022). Then, climate projections from regional climate models were used to determine the changes in temperature and precipitation in Baja California, which were further used to evaluate the impact of climate change on winegrape yields. Also, this study points out some limitations in data availability and the need for increasing monitoring and regional studies in grape and wine production for Baja California.

## 2. Data and Methods

### *2.1. Study Region*

The study was performed in the Mediterranean region of Northwestern Baja California in two viticulture areas (32.13° N–31.87° N and 116.60° W–116.20° W; 31.55° N–31.30° N and 116.40° W–116.16° W) shown in Figure 2, which include the main wine-producing regions of the state: Valle de Guadalupe and Ojos Negros (the new wine route) and Santo Tomas and San Vicente in the old wine route of Baja California. These valleys, located between 100 m and 530 m elevation above sea level, produce 99% of the winegrape production in Baja California in approximately 2160 ha [36].

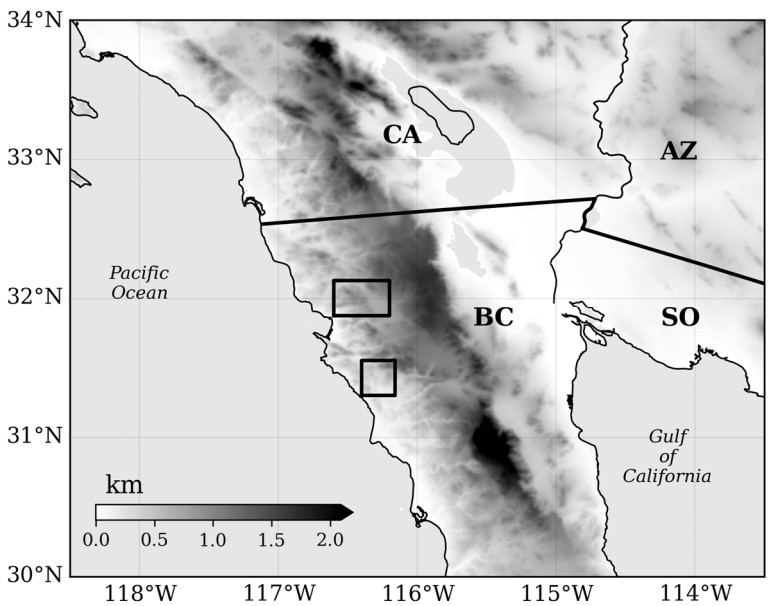

**Figure 2.** Baja California wine regions used in the analysis (black boxes). The gray color shows the topography of the study area in km.

The selected domains are characterized by a Mediterranean climate, with a mean annual temperature between 12 °C and 18 °C and accumulated precipitation between 100 mm and 300 mm [37]; maximum precipitation occurs during winter and is associated with the passage of frontal systems [38], the subtropical jet stream, and atmospheric rivers [39]. More than 40 winegrape varieties are grown in these valleys, of which Cabernet Sauvignon is the most representative for red wine (20% of the planted area in Baja California), with an average yield of 5 tons/ha. At the same time, Chenin Blanc is the main variety for whites (6.77% of the planted area), with a yield of 8 tons/ha [40].

### *2.2. Data*

#### 2.2.1. Climatic Data

Viticulture regions in Baja California State are characterized by a lack of in situ observations in recent years. Instead, in this work, available data from gridded observations and reanalysis databases are taken as an approximation to represent the climatic proxies, like other studies [1]. Monthly minimum (Tn), maximum (Tx), average temperature (T), and precipitation (Pre) data were used from the following databases:

Livneh. A gridded database that includes daily gauge observations with a spatial resolution of 1/16° (~6 km) for 1950–2013 [41]. It is available online at https://psl.noaa.gov/data/gridded/ (accessed on 10 July 2023). Livneh data are used as the reference for observations in this study. This data set was used in a recent study of climate change in wine regions in Mexico [1].

ERA5. Fifth generation of atmospheric reanalysis data from the Center for Medium-Range Weather Forecasts (ECMWF), which combines global numerical model forecast data with observations from around the world. The spatial resolution is 0.25° latitude and longitude. Data are available from 1979 to three months earlier in real time [42,43]. Available online at https://www.ecmwf.int/, accessed on 17 July 2023.

ERA-Interim. ERA-Interim reanalysis data [44] at a 3-hourly temporal resolution and approximately 75 km spatial resolution were used as boundary conditions to force the RCMs during the evaluation period, as explained in the next section. It is available at https://www.ecmwf.int (accessed on 17 July 2023).

### 2.2.2. Economic Data

The economic data used in this work consist of the real minimum wage for Baja California obtained from the National Commission of the Minimum Wages (CONASAMI), available online at https://www.gob.mx/conasami/documentos/tabla-de-salarios-minimos-generales-y-profesionales-por-areas-geograficas (accessed on 9 October 2023), and the price of the winegrape from the SIAP (available online at https://www.gob.mx/siap/acciones-y-programas/produccion-agricola-33119, accessed on 16 October 2023). Both wage and price are for the period from 1981 to 2022 and deflated to 2018 as the base year.

### 2.2.3. Climate Change Scenarios

For temperature and precipitation scenarios, daily outputs of the simulations from the only two regional climate models (RCMs: RegCM4 and RCA4) available for the Coordinated Regional Dynamical Experiment (CORDEX) Central America, Caribbean, and Mexico (CAM) domain (CORDEX-CAM) were used; these models have been evaluated for the studied region [1,45]. For historical evaluation, the RCMs were forced with ERA-Interim reanalysis data from 1981 to 2005 (reference period in this work), and the 21st century scenarios were forced by three general circulation model (GCM) outputs, the HadGEM2-ES, MPI-ES, and GFDL-ESM2M from the CMIP5 (Climate Model Intercomparison Project phase 5). The period of 2021–2040 is considered as the Near Future (NF), and 2041–2060 as the Intermediate Future (IF). The analysis was also performed for two different greenhouse gas emissions scenarios, the RCP2.6 (low emissions) and RCP8.5 (high emissions) radiative forcing scenarios. Data are available at https://esgf-data.dkrz.de/search/cordex-dkrz/ (accessed on 10 July 2023). A mean ensemble of the two model simulations was analyzed for the low- and high-emission scenarios.

RegCM4. Regional Climate Model version 4.7 [46]. This is an open-source regional model developed and updated by the Earth System Physics of the International Center for Theoretical Physics (ICTP) in Trieste, Italy. The hydrostatic version at 25 km (latitude longitude) of horizontal resolution is available for the CORDEX-CAM domain at https://gforge.ictp.it/gf/project/regcm/frs (accessed on 12 June 2022).

RCA4. The Rossby Center Regional Climate Model (RCA) is a regional model from the Swedish Meteorological and Hydrological Institute (SMHI; www.smhi.se, accessed on 11 December 2023). It is a hydrostatic model based on the numerical weather prediction model HIRLAM [47] conducted for the CORDEX domain [48,49] at a horizontal resolution of 50 km (latitude, longitude) spatial resolution.

HadGEM2-ES. Hadley Center Global Environment Model version 2 [50].
MPI-ES. Max Planck Institute Earth System Model [51].
GFDL-ESM2M. Geophysical Fluid Dynamics Laboratory Earth System Model [52].

### 2.3. Method
#### 2.3.1. Regional Climate

Seasonal temperature and winter accumulated precipitation (November to April) were calculated for each region in Figure 2. Then, an area-weighted average was applied to obtain time series for both regions as follows:

$$S = \frac{\sum_i^n \sum_j^m X_{ij} \cdot a_{ij}}{A} \qquad (1)$$

where $S$ represents the time series of each of the two domains for temperature or precipitation, the subscripts $i$ and $j$ are values that correspond to the latitude and longitude position of each grid point with sizes $n$ and $m$; $X_{ij}$ is the seasonal temperature or precipitation with dimensions in time, longitude, and latitude, $a_{ij}$ is the corresponding grid area, and $A$ is the total area for each winegrape region in Figure 2. Since the two regions are characterized by a Mediterranean climate, results in this work are also obtained as an average of the two regions as $(S_1 + S_2)/2$.

### 2.3.2. Regression Model

Two general linear regression models were constructed: one is used to estimate the relationship between the climate and economy on winegrape annual yield (Equation (2)), and the other model considers only the climatic effect on yield (Equation (3)). The linear functions integrate the main climatic influence and some economic determinants that approximate the expected winegrape yield in Baja California. Linear regression models have been shown to fit properly for modeling the response of the grape yield [53]. The growing season (April–October) is an important stage of the grape phenology that coincides with the growth and maturation of the berries and the harvest, whereas winter precipitation (November–April) favors water availability for the following cultivation cycle [8]. After testing the contribution of individual variables, the two winegrape yield models were set as follows:

$$Y = F(T_x,\ T_n,\ Pre,\ W,\ P) \qquad (2)$$

$$Y = F(T_x, T_n, Pre). \qquad (3)$$

In Equations (2) and (3), $Y$ represents the winegrape yield in function ($F$) of $T_x$ and $T_n$, which are the mean maximum and minimum temperatures during the winegrape growing season; $Pre$ is the antecedent winter precipitation. The economic proxies in Equation (2) are the annual minimum wage ($W$) and the grape price ($P$) for Baja California State during the study period. These two variables are the most representative of both production costs and market interactions. The labor cost represents more than 50% of the total operating costs, and the low grape price scheme influences the quality of the winegrape [9,15,54,55]. All variables are positive; before developing the regression models, the variables were normalized with their maximum value.

Including too many variables can easily lead to an inadequate prediction accuracy (e.g., variance overestimation), which makes interpretation of the model difficult [53,56]. For this reason, the predictor variables need to be appropriately selected in the development of the regression models. The most relevant variables for the crop were chosen by leading independent regressions between grape yield and climatic and economic variables, like other studies [57]. Afterward, multicollinearity tests were performed to avoid overfitting [56,58]. The regression models in Table 1 were obtained without detrending the annual data using the climate variables from Livneh (observations) and from the ERA5 reanalysis independently, to assess the skill of ERA5 relative to the observations.

The regression model results are used as a control base for the historical period, and then the same models are used to obtain the yield with the RCMs during the historical period forced with the GCMs and under climate change scenarios to obtain the future changes in the climatic variables. The regression models are run again to obtain approximate future yield and their possible changes.

**Table 1.** Winegrape yield (ton/ha) regression models for the Mediterranean Baja California region using mean seasonal climate information (Tx and Tn) from Livneh (obs) and ERA5 (reanalysis), and annual wages (W) and prices (P) of the winegrape production during 1981–2013. The regression models for the regional climate models (RegCM and RCA), forced with ERA-Interim, and the ensemble mean for the baseline (1981–2005) are also presented.

| Datasets | Equation—Climatic and Economic Variables |
|---|---|
| Livneh | $Y = 2.8 \times 10^4 - 1.5 \times 10^4 T_x - 6.6 \times 10^3 T_n + 2.3 \times 10^3 Pre + 491W - 2.7 \times 10^3 P$ |
| ERA5 | $Y = 5.6 \times 10^4 - 2.8 \times 10^4 T_x - 2.1 \times 10^3 T_n + 2.5 \times 10^3 Pre + 565W - 5.0 \times 10^3 P$ |
| RegCM-EI | $Y = 2.8 \times 10^4 - 1.7 \times 10^4 T_x - 3.3 \times 10^3 T_n + 1.9 \times 10^3 Pre + 432W - 2.9 \times 10^3 P$ |
| RCA-EI | $Y = 4.5 \times 10^4 - 2.6 \times 10^4 T_x - 2.5 \times 10^3 T_n + 1.9 \times 10^3 Pre + 432W - 2.9 \times 10^3 P$ |
| Ens-mean | $Y = 1.0 \times 10^4 - 981 T_x - 1.0 \times 10^4 T_n + 4.9 \times 10^6 Pre + 432W - 2.9 \times 10^3 P$ |
| | Equation—Climatic variables only |
| Livneh | $Y = 2.7 \times 10^4 - 1.5 \times 10^4 T_x - 6.6 \times 10^3 T_n + 2.2 \times 10^3 Pre$ |
| ERA5 | $Y = 5.4 \times 10^4 - 2.9 \times 10^4 T_x - 2.1 \times 10^3 T_n + 2.5 \times 10^3 Pre$ |
| RegCM-EI | $Y = 2.6 \times 10^4 - 1.7 \times 10^4 T_x - 3.2 \times 10^3 T_n + 1.9 \times 10^3 Pre$ |
| RCA-EI | $Y = 4.4 \times 10^4 - 2.6 \times 10^4 T_x - 1.2 \times 10^3 T_n + 2.5 \times 10^3 Pre$ |
| Ens-mean | $Y = -1.2 \times 10^4 - 981 T_x - 1.0 \times 10^3 T_n + 4.9 \times 10^6 Pre$ |

## 3. Results

First, we describe the climatic characteristics and winegrape yields for the two production areas in Section 3.1, and in Section 3.2, we present the climate projections of temperature and precipitation for the Near Future (NF) and Intermediate Future (IF) periods with the two CORDEX-CAM RCMs and the mean ensemble. In Section 3.3, we describe the possible impacts of the changes in maximum and minimum temperature and precipitation on the future winegrape yield in the region.

### 3.1. Climate Characteristics and Regression Models

Favorable conditions for winegrape production are characterized by a mean growing season temperature (GST) between 12° and 22 °C, e.g., [1]. Table 2 indicates that GST in Baja California varies between 17° and 20.2 °C according to Livneh's observations; meanwhile, ERA5 is approximately 1 °C warmer, but it is still suitable for viticulture. Mean maximum temperature of the growing season is 26.5 °C, and ERA5 slightly underestimates this value (25.9 °C). Both time series are well correlated, with positive trends; however, ERA5 shows a statistically significant trend (0.23 °C/decade; Table 2 and Figure 3). Mean minimum temperature from observations is 10.4 °C, whereas the reanalysis data has a strong positive bias of ~4 °C with a significant trend of 0.21 °C/decade, which is clearly seen in Figure 3. The winter season is characterized by an average of 364 mm of accumulated precipitation (Table 2, Figure 3b). The reanalysis presents a deficit of ~100 mm, but both data sets show similar trends (>20 mm/decade, not significant). It is important to mention that the period of Livneh's observations is nine years less than the reanalysis, and this might have implications for the difference in the statistical trends in ERA5, but not for the regression models, as we are also interested in the interannual variations.

**Table 2.** Descriptive statistics of the mean climate characteristics of the wine regions of Baja California during 1981–2013 using observations from Livneh (ERA5 values in parenthesis during 1981–2022). The Mean Bias Error (MBE) is the ERA5 bias relative to Livneh. Trends are calculated using the 'Sen' estimate of slope [59]. Statistical significance trends at the 95% confidence level are marked with an asterisk obtained with the Mann–Kendall test [60].

| Variable | Mean | MBE | Std | Min | Max | Trend/Decadal |
|---|---|---|---|---|---|---|
| Growing season maximum temperature (Tx, °C) | 26.5 (25.9) | −0.8 | 0.9 (0.7) | 24.8 (24.5) | 28.5 (27.6) | 0.10 (0.31 *) |
| Growing season average temperature (T, °C) | 18.3 (19.9) | 1.0 | 0.7 (0.6) | 16.8 (18.7) | 20.1 (21.4) | 0.04 (0.23 *) |
| Growing season minimum temperature (Tn, °C) | 10.4 (14.9) | 4.7 | 0.8 (0.5) | 8.4 (13.8) | 12.8 (16.2) | −0.06 (0.23 *) |
| Winter precipitation (Pre, mm/season) | 364.1 (261.4) | −97.2 | 194.9 (132.16) | 91.9 (69.6) | 852.2 (620.4) | −21.65 (−19.72) |

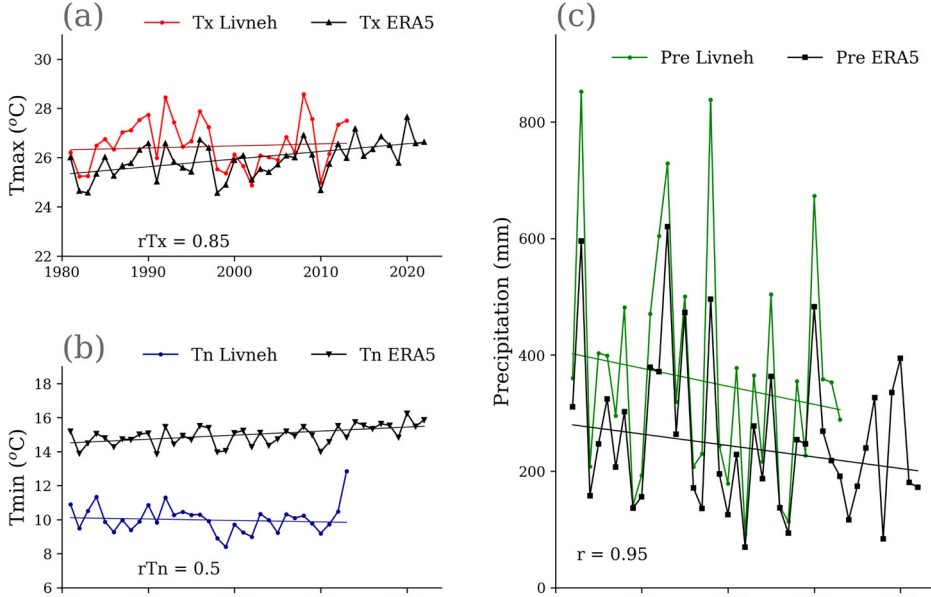

**Figure 3.** Growing season temperature and winter precipitation. (**a**) Maximum and (**b**) minimum temperatures (Tx, Tn) of the growing period (April–October) and (**c**) winter precipitation (November–April). The correlations (r) between Livneh's observations and ERA5 reanalysis time series are found below in the figure. All correlations are statistically significant with a confidence level of 95%.

Figure 4 shows the grape yield observations for Baja California and the results from the regression models, with (Figure 4(left)) and without (Figure 4(right)) considering the economic variables. During the first two decades of the yield observations, the variability in the winegrape yield was high in comparison to the rest of the period, and afterward, a negative trend is observed.

The time series from the fitted regression models based on climatic and economic variables show that the explained yield variance with ERA5 is larger (r = 0.53, $R^2$ = 28%) than that using Livneh (r = 0.42, $R^2$ = 18%); the root mean square error (RMSE) is similar in the two regressions, ~2 ton/ha (Figure 4). Not considering the economic variables in the regression models results in slightly smaller correlation coefficients and variance explained ($R^2$ = 18% with ERA5 vs. 14% with Livneh). Although the correlation is low in both cases, the grape yield observations are adequately reproduced, with the reanalysis model regression producing a slightly better representation of the yield than observations from Livneh, especially considering the model with the economic regressors.

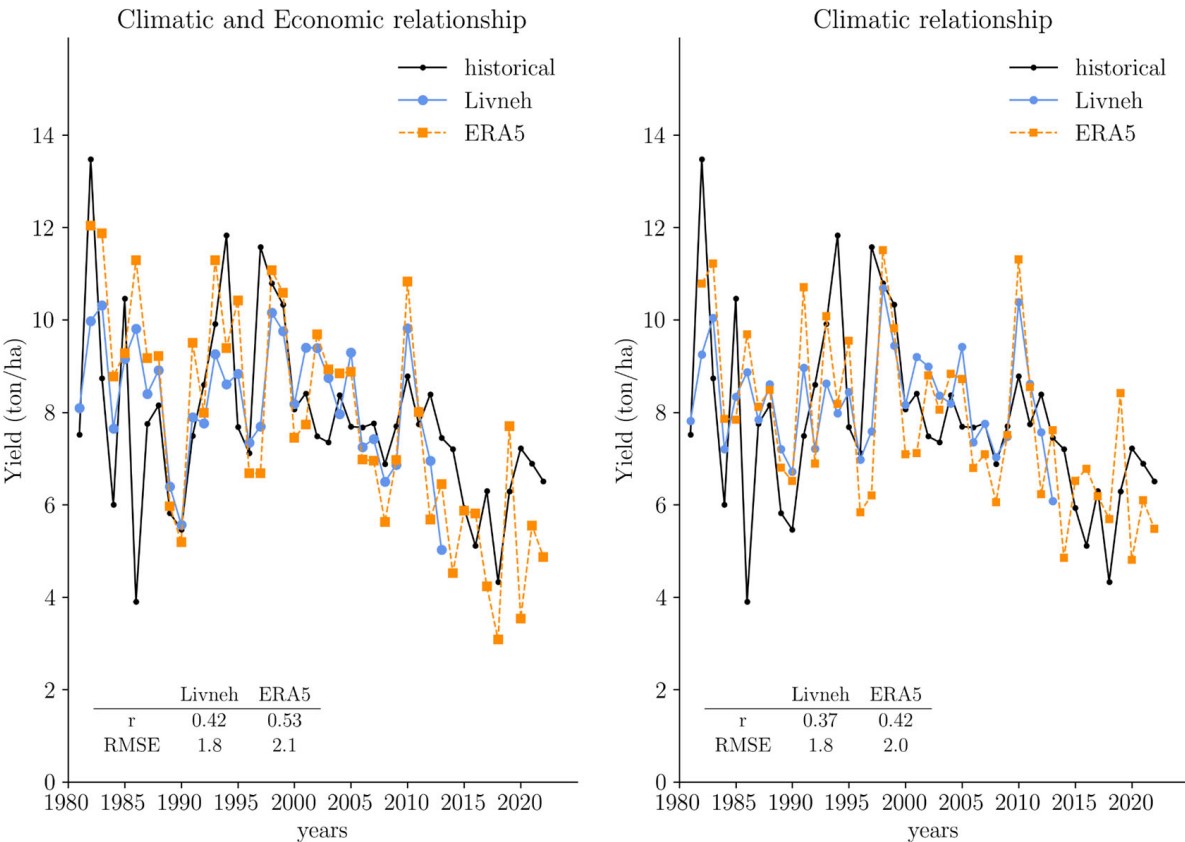

**Figure 4.** Annual winegrape yield for Baja California wine region. The black line represents the observations, the blue line is for the regression model using Livneh's observations (1981–2013), and the orange one is for the ERA5 (1981–2022). Regression models with economic and climatic variables (**left**), and regression models with climatic variables only (**right**). The correlations (r) between the observation and reanalysis series and the root mean square error (RMSE) are found below in the lower part of the figure.

Also, regression models were set for the RCMs forced with ERA-Interim and the ensemble mean (forced with the GCMs) in the baseline period (Figure 5). The correlation coefficients in this example were lower, and the RMSE remained similar in comparison with Livneh and ERA5 models in Figure 4, possibly because ERA-Interim has a much lower spatial resolution than ERA5 (75 km vs. 32 km approximately). Although the RCMs capture the historical mean yield, the models do not simulate the interannual variability in the winegrape yield well. It is important to highlight that in Figure 5(left), the winegrape yield is better represented when the economic variables are added to the model. For example, at the end of the 1980s, the models reproduced a drop in grape yield, as observed in the historical data. Moreover, it is important to clarify that the *ens* of the GCMs is not expected to capture the interannual variation, as the chronology of the GCMs does not correspond to those of the observations. This time series was added to the figures to see the simulated mean yield and its trend relative to the RCMs forced with ERA-Interim, e.g., [1].

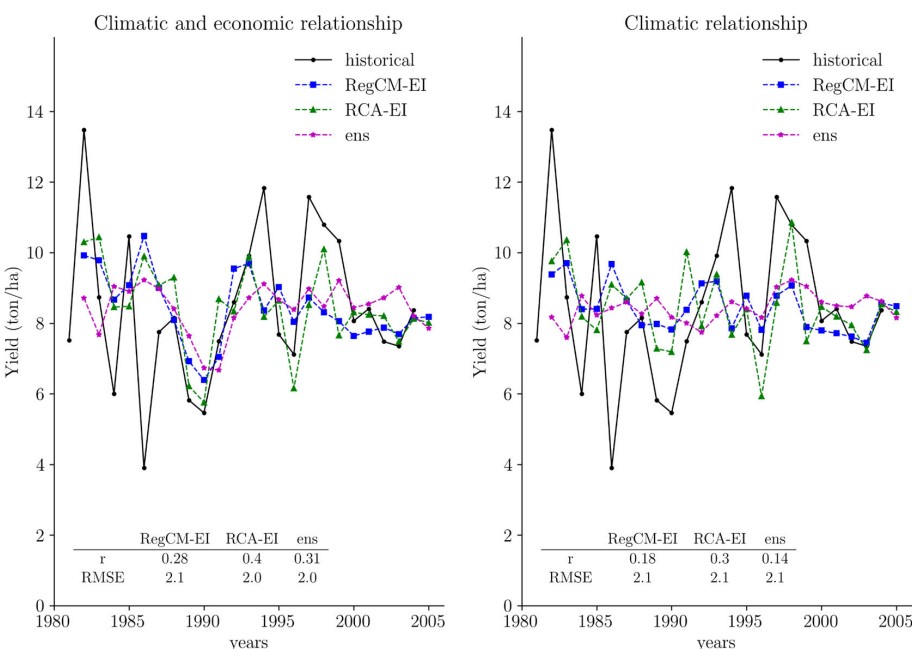

**Figure 5.** As in Figure 4, but for the RegCM (blue line) and RCA (green line) forced with ERA-Interim (EI) and with the ensemble (ens) mean (magenta line) forced with the GCMs in the baseline period.

### 3.2. Seasonal Climatic Conditions with the Ensemble of RCMs

The results show that the RCM mean ensemble (RegCM4 and RCA4) forced by the GCMs reproduces the general pattern of the annual cycle of minimum and maximum temperature (Figure 6a) of the winegrape regions studied in comparison with ERA5 and observations. However, as it was seen in Table 2 and Figure 2 for ERA5 reanalysis, the RCMs also overestimate minimum temperature, whereas maximum temperature is overestimated only in the summer months. Although the ensemble mean of the RCMs was also able to capture the general pattern of the annual cycle of precipitation, at the end of the winegrape growing season, it shows a deficit in winter (the main rainy season) and an overestimation at the end of the summer (Figure 6b).

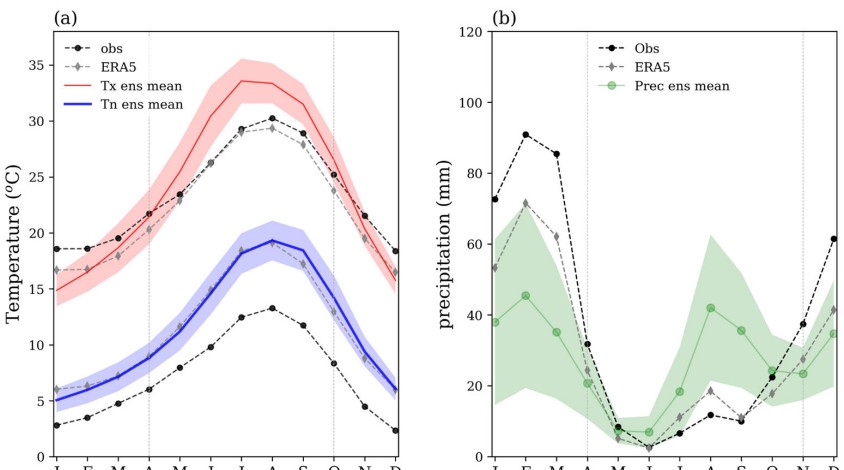

**Figure 6.** Annual cycle of (**a**) maximum and minimum temperature, and (**b**) precipitation for the 1981–2005 period for Livneh observations, ERA5 reanalysis, and the mean ensemble of the models forced by the GCMs. The solid-colored lines represent the ensemble mean, and the shaded areas indicate ±1.5 standard deviations ($\sigma$) across the ensemble member. The winegrape growing season (April–October) and the winter season (November–April) are marked with gray vertical lines in (**a**) and (**b**), respectively.

### 3.3. Regional Climate Change Scenarios

The climate change projections were assessed for the RCP2.6 and RCP8.5 scenarios using the mean ensemble of the six simulations for the NF (2021–2040) and IF (2021–2060) periods. Relative to the baseline period (1981–2005), the ensemble mean shows an increase in the annual cycle of the maximum (Tx) and minimum (Tn) temperatures (Figure 7a,b) but differs in intensity by scenario and period (NF or IF). In the most severe scenario (RCP8.5) during the IF, both $T_x$ and $T_n$ are ~2 °C above the baseline and up to ~3 °C warmer from August to October, particularly $T_n$. Consistent with our results, similar conditions have been found in our domain and surroundings using GCMs [61–63] and with RCMs [1].

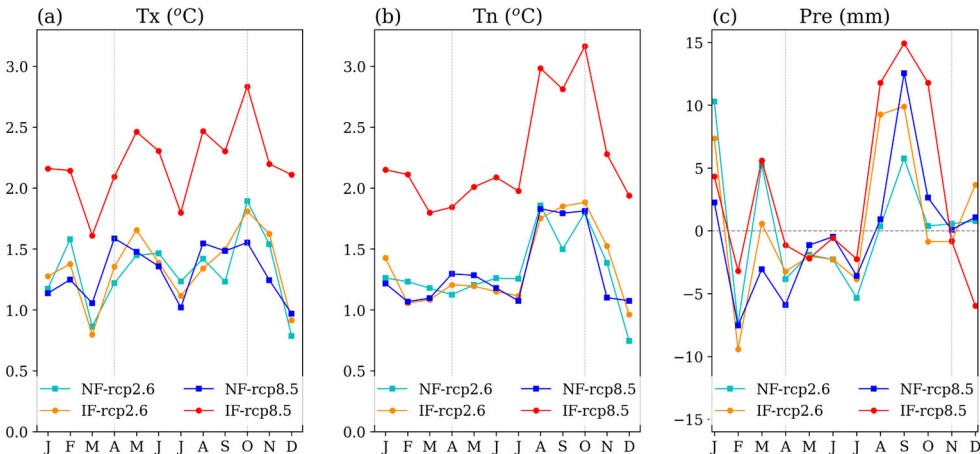

**Figure 7.** Monthly changes in the ensemble projections for the NF and IF relative to the ensemble mean during the evaluation period (1981–2005) for (**a**) maximum temperature (Tx), (**b**) minimum temperature (Tn), and (**c**) precipitation (Pre). The growing season (April–October) is marked with gray vertical lines in (**a**,**b**) and the winter season (November–April) in (**c**).

The growing season could be affected by warmer conditions in the two periods and under the two scenarios, with increments of $T_n$ and $T_x$ by ~1 °C from April to July (from vine budburst to inflorescence). A larger increment (~2 °C) in $T_x$ is projected for the IF-RCP8.5 scenario, and $T_n$ could increase up to 3.5 °C during the harvest season (August–October). The effects of warmer conditions on winegrape production are discussed in the following sections.

The ensemble projections suggest that regional winter precipitation changes are mostly negative on average, but more uncertain during the harvest season when precipitation is projected to increase under the two scenarios and periods (Figure 7c). The scenario with the largest changes in precipitation is the NF-RCP8.5, which has a deficit of −12.4 mm in the winter season, and during the summer, there is a possible increase of 5 mm and up to 15 mm in the IF-RCP8.5 scenario. The typical rainy season shows more future uncertainty than the summer season; the different greenhouse scenarios project an increase in precipitation in January, with a decrease in February and at the end of the season, with differences that range from −5 mm/month to 5 mm/month. The precipitation uncertainty in this region was also observed by Cavazos and Arriaga-Ramirez (2012) [64] using a single GCM (Had-GEM2), possibly due to uncertainties in the changes of the position of the subtropical jet stream [65].

### 3.4. Winegrape Yield According to Climate Change Scenarios

Possible changes in winegrape yield were projected for the NF and the IF using the seasonal changes in temperature and precipitation for the two scenarios (shown in Table 3). The possible changes in temperature and precipitation of the NF and IF scenarios are added to the regression models based on Livneh and ERA5, to produce other possibilities of changes in winegrape production. The climate and economic model using the observed

Livned data shows a possible grape yield reduction of ~21%, except in the IF-RCP8.5, which indicates a larger reduction of 35% yield. The effect of not considering economic variables, the climate -only model, suggests a persistent, but slightly smaller yield reduction (~30% for IF-RCP8.5). This effect is also seen in the ERA5 regression model. Evaluations made using ERA5 reanalysis as the reference for the climatic information show a much larger (almost double) negative impact on grape yield than Livneh's models, which reflect the large biases of the surface climate variables of ERA5 in the study region, particularly the overestimation of $T_n$ (Table 2 and Figure 3) and the RCMs (Figure 6a). These regression results suggest possible uncertainties not only according to the future climate scenarios, but also on the observational data.

**Table 3.** Changes in winegrape yield (%) in the NF and IF under the two RCP scenarios according to the climate and economic and the climate only regression models. Percentage changes of the ensemble mean were obtained relative to the observed average yield from the linear fitting using Livneh (1981–2013) and ERA5 (1981–2022) climate variables as the historical average.

| Data | Scenarios | Climate and Economic | | | Climate Only | | |
|---|---|---|---|---|---|---|---|
| | | Historical Yield | NF | IF | Historical Yield | NF | IF |
| Livneh | RCP 2.6 | 8.25 | −20.8% | −21.6% | 8.25 | −18.2% | −18.8% |
| | RCP 8.5 | ton/ha | −21.9% | −35.1% | ton/ha | −19.1% | −30.6% |
| ERA5 | RCP 2.6 | 7.81 | −43.8% | −45.7% | 7.81 | −42.7% | −44.3% |
| | RCP 8.5 | ton/ha | −46.3% | −78.6% | ton/ha | −44.8% | −72.4% |

## 4. Discussion and Conclusions

In this study, the possible impact of climate change on the winegrape yield was evaluated for the major winemaking regions in Baja California, Mexico, in the municipality of Ensenada, where more than 80% of the wine in Mexico is produced. Moreover, this wine region is the second largest winegrape producer in Mexico after Zacatecas state [1]. To assess the role of climate and economic factors, we constructed two linear regression models, one considering the role of the mean surface climate conditions and some economic variables and another with only climatic variables. We used the mean ensemble simulation from the only two RCMs (forced with three GCMs) available for CORDEX-CAM for the historical period and two future periods under low and high emission scenarios. Castillo et al. (2023) [1] uses the same numerical models to analyze present and future climatic changes in winegrape regions in Mexico. However, our study also includes the possible role of economic variables and regression models to simulate winegrape yield production, something that has not been done before in Mexico.

The regional climate analysis in this work and other results support that during the study period (1981–2022), the main viticultural regions in Baja California are characterized by positive trends in temperature (e.g., [45,66,67]) and negative trends in precipitation (e.g., [45,67]), albeit not statistically significant. In the same period, the winegrape yield observations reveal a reduction of 52% in recent decades. Using environmental elementary variables for the appropriate development of the winegrape, such as temperature (Tx and Tn) and precipitation, the mean yield and observed trends in the studied region were adequately reproduced by the regression models. Our study shows that considering the appropriate economic indicators, the variance in the yield observations can be better represented, since the correlation between the linear model and observations increased when wage and winegrape price were considered.

The climate change projections by the RCMs [1,68] and global models [61,62] for future periods indicate agreement with warmer projected conditions in our study region. The mean ensemble of the RCMs projects an increase of at least 1 °C in the NF. This could affect the initial vegetation development period in grapevines, particularly budburst and inflorescence. In these stages, environmental temperature plays a crucial role in vine

physiology [8,24]. Temperatures exceeding a specific optimum cultivar heat can induce a significant decrease in grapevine photosynthetic productivity [69,70].

In the maturation and harvest stage, the climate scenarios suggest an increment of 1.5 °C to 3 °C in the most severe scenario. More elevated temperatures during *véraison* can increase grape sugar accumulation, reduce grape acidity, and accelerate the development and maturity of the vine fruit [8,24,71,72] and other negative impacts that can lead to lower production [8,73,74] or grape quality [75,76]. In addition, favorable areas for growing grapevines in Baja California could be reduced under climate change scenarios, particularly at the end of the 21st century under RCP8.5 [1,77], as it has been projected for other viticulture regions, e.g., [78], including southern California [77].

The changes in the projected winter precipitation can vary from month to month, but this uncertainty can carry to hot and dry atmospheric conditions that may also result in negative impacts even in irrigated vines [74,76,79]. On the other hand, when vines are irrigated appropriately during winter, it may be beneficial, but high humidity may increase the likelihood of diseases [35,80].

The expected climate changes, particularly higher temperatures, in the NF and IF imply that grape production in the study region might not be optimal. The regression models reveal that grape productivity could experience average annual reductions between ~18% and 20% (NF-RCP2.6 and IF-RCP2.6), as in Fraga et al. (2020) [81]. As temperature is one of the two relevant climatic factors in grape productivity, it is not surprising that results in RCP8.5 scenarios showed a dramatic drop in grape productivity in both regression models.

The results showed that grape yield is also sensitive to socio-economic factors, supported by the climate and economic regression model, which indicates that in combination with non-favorable (warmer and drier) climatic conditions for grapevine phenology, productivity may decrease at least ~3% more (in all the cases) in comparison with the scenarios considering climatic only. This suggests that the regional economy also plays a role in grape productivity in Baja California, in accordance with other works, such as González Andrade (2015) [55], and other regions [9]. These studies suggest that temperature and precipitation changes will produce variations in terms of the operating cost paid by producers.

The results described here are based on the current winegrape yield and varieties cultivated in Baja California, but we did not differentiate yield according to variety in our study. Therefore, it is important to initiate regional studies on winegrape adaptation to climate change to mitigate climate change impacts in the wine sector. The sustainability of wine producers will depend on the resilience of local agriculture [82] and adaptive mitigation strategies, for instance, including more drought-tolerant crops by cultivar or variety selection [83,84]. Efficient irrigation systems should be studied [84], as well as dry-farming techniques [77] and suitable areas at higher elevations [77,85].

In summary, consistent with other studies, temperature is rising in the Baja California wine-making regions, and future climate projections suggest that warming conditions will continue under low- and high-emission scenarios. With this temperature tendency, it is expected that winegrape production will decrease, as our regression models suggest. Therefore, it is important to increase monitoring of climate and economic variables and to continue researching adaptation strategies to mitigate the possible effects of climate change. Our study indicates that it is also important to consider the economic effects in adaptation initiatives.

**Author Contributions:** Conceptualization, M.H.G., M.C.G.-L., T.C. and I.E.; methodology, M.H.G., M.C.G.-L. and T.C.; validation, M.H.G.; formal analysis, M.H.G.; investigation, M.H.G., M.C.G.-L. and T.C.; data curation, M.H.G.; writing—original draft preparation, M.H.G.; writing—review and editing, M.C.G.-L., T.C. and I.E.; supervision, M.C.G.-L. and T.C.; funding acquisition, M.C.G.-L., I.E. and T.C. All authors have read and agreed to the published version of the manuscript.

**Funding:** This research was funded by the National Council of Science and Technology (CONACYT) grant no. 770216.

**Data Availability Statement:** The data presented in this study are available in: The reanalysis data of ERA5 can be obtained at https://www.ecmwf.int/ (accessed on 17 July 2023); CORDEX-CAM at https://esgf-data.dkrz.de/search/cordex-dkrz/ (accessed on 10 July 2023). Livneh at https://psl.noaa.gov/data/gridded/ (accessed on 10 July 2023); wage for Baja California at https://www.gob.mx/conasami/documentos/tabla-de-salarios-minimos-generales-y-profesionales-por-areas-geograficas (accessed on 9 October 2023), and the price of the winegrape available online at https://www.gob.mx/siap/acciones-y-programas/produccion-agricola-33119 (accessed on 16 October 2023). All the data used in this work were cited within the article.

**Acknowledgments:** We thank CONACYT for the Ph.D. Scholarship Grant awarded to the first author for her Ph.D. studies at Universidad Autónoma de Baja California (UABC-Mexico). We acknowledge the support from the CICESE Physical Oceanography Department and Rio Arronte Foundation (project A-447). We also thank the anonymous reviewers for their comments.

**Conflicts of Interest:** The authors declare no conflicts of interest.

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
