# Peer review of "Impacts of Climate Change in Baja California Winegrape Yield"

_climate, doi:10.3390/cli12020014_

Round 1
Reviewer 1 Report
Comments and Suggestions for Authors
Please complete the discussion section
Comments on the Quality of English Language请完成讨论部分
Author Response
Dear reviewer,
We appreciate the time and effort dedicated to providing your constructive feedback on our work. We have made changes to the manuscript in the discussion section to incorporate the suggestion made in your review. The changes are highlighted in the original text. We thank you again for taking the time to review our manuscript.
Sincerely,
Marilina Hernandez Garcia
Instituto de Investigación Oceanologica
Universidad Autónoma de Baja California
Reviewer 2 Report
Comments and Suggestions for Authors
The manuscript is dedicated to climate change impacts on agriculture: wine grape yield in Baja California, Mexico. Climate change contribute in reducing of regional winegrape producing. The authors assess perspectives using climatic and economic variables.
The primary data includes best available climatic and economic information, including 1979-2023 reanalysis, the price of the winegrape 1981-2022 and two regional climate models. The period of 2021-2040 is considered as the Near Future (NF) and 2041-2060 as the Intermediate Future (IF), and RCP2.6 (low 158 emissions) and RCP8.5 (high emissions) radiative forcings scenarios have been considered.
Two general linear regression models were constructed, one is used to estimate the relationship between the climate and economy on winegrape annual yield (eq. 2), and the other model considers only the climatic effect on yield.
The growing season could be affected by warmer conditions in the two periods and under the two scenarios, with increments of Tn and Tx by ~1ºC from April to July (from vine budburst to inflorescence). A larger increment (~2ºC) in Tx is projected for the IF RCP8.5 scenario, while Tn could increase up to 3.5°C during the harvest season (Aug-Oct). The ensemble projections suggest that regional winter precipitation changes are mostly negative on average, but are more uncertain during the harvest season when precipitation is projected to increase under the two scenarios and periods.
The climate-economic model based on observed data shows a possible grape yield reduction of ~21% except in the IF-RCP8.5, which indicates a reduction of 35% yield. The effect of not considering economic variables, the climatic-only model suggests a persistent, but slightly less yield reduction (~30% for IF-RCP8.5). This effect is also seen in the ERA5 regression model.
As concluded: the sustainability of wine producers under climate change will depend on the resilience of local agriculture and implementation of adaptive strategies, for instance, including more drought-tolerant crops by cultivar or varieties selection, and efficient irrigation systems should be studied.
Author Response
Dear reviewer,
We appreciate the time and effort dedicated to providing your constructive feedback on our work. We have made changes to the manuscript to incorporate all the suggestions made for the reviewers. The changes are highlighted in the original text. We thank you again for taking the time to review our manuscript.
Sincerely,
Marilina Hernandez Garcia
Instituto de Investigación Oceanologica
Universidad Autónoma de Baja California
Reviewer 3 Report
Comments and Suggestions for Authors
General comments:
The manuscript proposed by Marilina Hernandez Garcia et al. discuss climate change scenarios and their possible impacts on winegrape yield. The research indicates that it is important to study on adaptation strategies to mitigate the possible effects of climate change, and indicates that it is also important to consider the economic effects in the adaptation initiatives.The topic chosen for this paper is highly meaningful.
1) The subject matter and scientific merit of the manuscript original is enough to warrant publication.
2) There are no logical errors in the presentation and in the approaches chosen.
3) The conclusions are valid and derived logically. The manuscript has some innovation.
4) The length of the manuscript is not appropriate, and it is suggested to simplify the description of “Introduction” and “Conclusions and discussion”.
5) The similarity between graphics is too large.The figures are mostly line charts.
I believe the research findings have some value, it is suitable for publication in "Climate".
Specific comments:
1、 The gridded observations and reanalysis databases have different start and end times. Can you tell me how to solve this problem when building Linear regression models?
2、 There are many regional climate models, why choose RegCM4 and RCA4?
3、 There are differences in the highest, lowest, average temperatures, and precipitation between Livneh's observations and ERA5 data, which are worth considering when establishing regression equations. ( lines 243 to 257).
4、 The RCMs overestimate minimum temperature, maximum temperature is overestimated only in the summer months,and precipitation shows a deficit in winter (the main rainy season) and an overestimation at the end of the summer? Since there are many differences in the historical climate simulated with the ensemble of RCMs, how reliable is it in simulating future climate? (lines 312 to 319).
5、 If possible, please consider whether there is a possibility of environmental adaptability for grape growth in Baja California.
Author Response
Dear reviewer,
We appreciate the time and effort dedicated to providing your constructive feedback on our work. We have made changes to the manuscript to incorporate the suggestions made in your review. The changes are highlighted in the original text and in this document. Our response to your comments are written in blue in this document to make them easier to read. We thank you again for taking the time to review our manuscript.
Sincerely,
Marilina Hernandez Garcia
Instituto de Investigación Oceanologica
Universidad Autónoma de Baja California

Reviewer 4 Report
Comments and Suggestions for Authors
This manuscript delves into the potential impacts of climate change scenarios on winegrape yield in Baja California, Mexico's foremost wine-producing region. Employing linear regression models, it predicts current yield by considering both climatic and economic variables.While the topic is undoubtedly of high interest and the approach of integrating climatic and economic variables is commendable, there are several critical issues that significantly hinder the manuscript's effectiveness and scientific robustness. Therefore, after a thorough evaluation, I regret to inform you that I recommend rejection of the manuscript in its current form. These issues are as follows:
Abstract:
The analysis of economic factors and their interplay with climatic variables requires greater elaboration for a more comprehensive understanding.
Introduction:
The transition from discussing the importance of the wine industry to the impacts of climate change needs to be more naturally and logically integrated.
Data and Methods:
The manuscript relies on outdated RCP scenarios from CMIP5, overlooking the newer SSP scenarios from CMIP6.
Results:
1. Recommendation: It is recommended that the boxes "a" and "b" in Figures 4 and 5 be removed to improve clarity.
2. Inconsistent use of the term 'RCP' and 'RMSE'.
3. Figure 3's inconsistent y-axis formatting across subgraphs and Table 3's non-compliance with standard three-line table format requirements .
Discussion and Conclusions:
1. There are formatting errors, such as in the section title ('3.Discussion and conclusions' should be '4. Discussion and Conclusions').
2. The discussion needs a broader comparative analysis with other studies and a more in-depth exploration of uncertainties in the results, including model limitations and climate prediction uncertainties.
Comments on the Quality of English LanguageUpon reviewing your manuscript on the impacts of climate change on winegrape yield in Baja California, I found the English language quality to be generally good. However, minor refinements could further clarify and polish the text. A brief review by a native English speaker or a professional editing service could be beneficial to fine-tune the language for enhanced readability and precision.
Author Response

(The authors gave the same response as above.)

Round 2
Reviewer 4 Report
Comments and Suggestions for Authors
This manuscript analyzed climate change scenarios and their possible impacts on winegrape yield inBaja California, the leading wine producer in Mexico. Linear regression models were used to predict the current yield based on climate and economic variables.this study points out some limitations in data availability and the need for rigorous regional studies in grape and wine production for Baja California. In general, the manuscript is well written. The literature is extensively reviewed, the methodology is explicitly described in the manuscript, and the results are logically interpreted. The content of the manuscript is relevant to the journal and is expressed logically. The author has been revised one by one in accordance with the amendments. I think it is acceptable to publish.